# The Potential Application of *Ecklonia cava* Extract in Scalp Protection

**Hayeon Kim** , **Hyunju Woo, Seoungwoo Shin** , **Deokhoon Park and Eunsun Jung** *

Biospectrum Life Science Institute, U-TOWER 18th FL, 767, Sinsu Ro, Suji Gu, Yongin City, Gyunggi Do 16827, Korea; bioyu@biospectrum.com (H.K.); biour@biospectrum.com (H.W.); biost@biospectrum.com (S.S.); pdh@biospectrum.com (D.P.)

* Correspondence: bioso@biospectrum.com; Tel.: +82-31-698-3122; Fax: +82-31-698-3123

**Abstract:** The scalp is exposed to environmental hazards including airborne pollutants, which exert adverse effects on skin health. Therefore, compounds for defending skin from pollutants have attracted interest in the cosmeceutical community. We investigated whether *Ecklonia cava* exhibited prophylactic effects against urban pollutants by measuring cell viability and cell cycle distribution in human follicle dermal papilla cells (HFDPC). The effect of *E. cava* on pollutant-induced damage to skin barrier was determined by measuring filaggrin and MMP-1 expression in both keratinocytes and in a skin explant model. In a clinical trial, the effect of *E. cava* on scalp skin of patients with scalp scale was observed by evaluating hydration and redness after 4 weeks of daily treatment with a shampoo containing *E. cava* extract. *E. cava* extract recovered the loss of cell viability and abnormal cell cycle distribution induced by urban pollutants in HFDPCs. It also attenuated pollutant-induced damage to skin barrier by decreasing MMP-1 and increasing filaggrin expression in keratinocytes and the epidermis of skin explants. Moreover, *E. cava* showed soothing effects on human scalp by increasing hydration and decreasing redness in a clinical trial. Collectively, *E. cava* extract may be a good candidate for therapeutic applications designed to repair or protect hair scalp.

**Keywords:** *Ecklonia cava*; scalp; urban dust; hydration; anti-inflammation

## 1. Introduction

The skin that grows hair from the top of the face to the nape of the neck is called the scalp, and it contains hair follicles, sebaceous glands and sweat glands. External stressors of which there may be many in air pollution, can disrupt healthy scalp condition and cause problems such as inflammation and hair loss [1,2].

Fine dust is any particulate matter that floats for a long time in the air and is less than 10 μm in diameter. It may occur in the form of nitrogen compounds, sulphates, ammonium or carbon compounds and all are known to have significant effects on human health [3,4]. Particulates can be produced by natural causes such as forest fires and volcanic ash, however, the major contributing factor is due to the burning of fossil fuels such as seen in car fumes, coal and oil by man-made causes [5]. In this study, urban dust (Standard Reference Material® 1649b, NIST) collected for more than 12 months in Washington DC by using a special filter was purchased and used as a source of pollutants containing polycyclic aromatic hydrocarbons (PAHs) from all particulate matters in the atmosphere.

Urban dusts directly affect human health, and WHO has designated ultra-fine particles of less than 2.5 μm in diameter as a first-class carcinogen [6]. Various studies have been reported on the adverse effect of air pollutants in skin. When the aryl hydrocarbon receptor (AhR) within skin cells is activated by environmental pollutants, it migrates from the cytoplasm to the nucleus and binds to the xenobiotic response element (XREs) [7]. Reactive oxygen species (ROS), inflammatory cytokines and matrix

metalloproteinases (MMPs) are increased by AhR activation, which leads to alterations in the cell cycle, inflammation, and degradation of extracellular matrix (ECM) [8,9]. In addition, pollutant-induced COX expression and filaggrin downregulation provoke skin barrier dysfunction [10]. In barrier-disrupted skin, pollutants can easily penetrate the skin and aggravate cutaneous inflammatory responses through ROS production [11]. The effects of pollutants on skin barrier and ECM homeostasis have been widely reported but effects on hair follicles and scalp skin have been poorly characterized. Ramos PM has reported that AhR overexpression in the nucleus was observed in miniaturized hair follicles in female pattern hair loss [12].

*Ecklonia cava*, also known as 'sea trempet', is a marine brown alga species found off the coast of Japan and Korea. It is known to contain an abundance of bioactive compounds such as phlorotannin, diecktol and fucoidan and therefore, has potentially useful pharmacological properties and may exhibit anti-oxidative, anti-inflammatory, anti-microbial, and anti-cancer effects [13]. An earlier study reported that *E. cava* can promote hair growth [14] however, its effects on scalp conditions or skin irritation induced by urban pollutants have not been investigated. Therefore, we designed this study to test the efficacy of *Ecklonia cava* extract in providing scalp protection.

## 2. Materials and Methods

### 2.1. E. cava Extraction

*Ecklonia cava* was washed and dried and then extracted with water at 100 °C for 2 h. This solution was passed through a 5 μm filter, concentrated with a decompression condenser, and then dried by lyophilization to form powder for further study.

### 2.2. Cell Culture

Human follicular dermal papilla cells (HFDPC) were obtained from Promocell (Heidelberg, Germany) and maintained in supplemented follicle dermal papilla cell growth medium (Promocell, Germany) at 37 °C, under 5% $CO_2$ atmosphere. HaCaT, a human keratinocyte cell line (CLS, Germany), was maintained in Dulbecco's Modified Eagle's Medium (DMEM), containing 10% fetal bovine serum (FBS) and 1% penicillin/streptomycin (all from Welgene, Korea) at 37 °C, under 5% $CO_2$ atmosphere.

### 2.3. Reagents

As ambient particular matter, we purchased Standard Reference Material®1649b (NIST) containing crucial components of urban dust like polycyclic aromatic hydrocarbons (PAHs) and used this in all experiments. MTT (3-(4,5-dimethylthiazol-2-yl)-2,5-diphenyl tetrazolium bromide) and propidium iodide were purchased from Sigma-Aldrich (St. Louis, MO, USA).

### 2.4. Evaluation of Cytotoxicity of HFDPC

Cytotoxicity of urban dust SRM-1649b on HFDPC was detected by MTT assay. Briefly, $2.5 \times 10^4$ cells were seeded in each well of a 24-well plate and incubated at 37 °C in humidified atmosphere of 5% $CO_2$/95% air for 24 h. SRM-1649b was added at various concentration into each well and incubated for an additional 72 h. MTT solution (50 μL, 1 mg/mL) was then added to all wells and allowed to form violet formazan in the cells for 4 h. After removing the media, cells were lysed with 150 μL of DMSO and 100 μL of lysate was aliquoted into a 96-well plate in order to determine cell proliferation by measuring the amount of formazan produced from MTT. Measurement was performed in a microplate reader at 570 nm and normalized against non-treated samples.

### 2.5. Cell Cycle Analysis

HFDPC were seeded into 6-well plates at $1.2 \times 10^5$ per well. Cells were treated SRM-1649b in the presence or absence of *E. cava* extract in each well for 48 h. After treatment, cells were collected and fixed with 70% EtOH overnight and washed with DPBS (Welgene, Korea) containing 2% FBS.

Propidium iodide-staining was performed to determine cell cycle distribution and analyzed with BD FACSCalibur™, flow cytometry, and BD CellQuest™ Pro Software (Becton Dickinson, San Jose, CA, USA).

### 2.6. Western Blotting

HaCaT, human keratinocytes, were treated with SRM-1649b in presence or absence of *E. cava* extract for 48 h or 72 h. Following treatment, the cells were washed with cold PBS and lysed with Pro-prep™ protein extraction solution (iNtRON biotechnology). Quantification of protein was by BCA assay. Protein samples of equal quantity were loaded and then separated on NuPAGE™ 4–12% Bis-Tris Protein Gels (Thermo Scientific, Waltham, MA, USA) and transferred to PVDF membrane. The following antibodies were used for detection in western blotting: anti-filaggrin (Biolegend, PRB-417P), anti-β-actin (Santa Cruz, sc-47778), and anti-MMP-1 (Santa Cruz, sc-58377). Antibody binding on the membrane was visualized by ImageQuant™ LAS 500 (GE Healthcare Life Sciences). Each protein band was analyzed using image J and the protein levels normalized to β-actin on the same blot.

### 2.7. Ex Vivo Study

To evaluate the protective activity of *E. cava* extract after exposure to pollution stress on living human skin explants, an ex-vivo study was conducted in laboratory BIO-EC (France). Skin explants were kept in BIO-EC's explants medium at 37 °C in humidified atmosphere of 5% $CO_2$/95% air. On day 5, explants were treated with a paper disk soaked with pollutant solutions. The pollutants used were as follows: Solution ICP (multi-elements standard V, mixture of heavy metals, Merck), benzene, anthrancene, xylene, toluene, benzopyrene, and naphtalene. On days 1, 2, and 5, 0.05% *E. cava* extract solutions were applied at the dose of 2 mg/cm$^2$ on the explants and spread using a spatula. After treatment, each skin explant was cut, fixed and paraffinized for the following steps. Filaggrin or MMP-1 immunostaining were performed with each monoclonal antibody. The nuclei were post-stained using propidium iodide. Each stained skin explant was evaluated with image J program and the data shown graphically.

### 2.8. Clinical Study

To evaluate the effects of *E. cava* extract on scalp, scalp redness, and hydration were analyzed at the Dermapro Skin Research Center (Seoul, Korea). Ethical approval for this study was obtained from the Ethics Committee of Dermapro. Volunteers aged from 24 to 54 (average: 44.7± 7.8, n = 21) with scalp scale and redness participated, and all provided written informed consent before the study. The testing product was manufactured as shampoo containing 0.02% of *E. cava* extract and was distributed to study subjects. It was recommended they use the shampoo once a day and massage into the scalp for two minutes. Scalp hydration on frontal and parietal region were determined by DermaLab (Cortex, Denmark), which is known to measure the water binding capacity of the stratum corneum based on the conductance measurement principle. Scalp scale and redness were taken using Aroma TS® and the score of redness was assessed by two researchers according to criteria for scalp redness status (Table 1). Scalp scale, redness and hydration were evaluated at baseline and after 4 weeks of treatment. Statistical analysis of the clinical test was conducted using the SPSS (R) software program (IBM, Armonk, NY, USA).

**Table 1.** Photo assessment for scalp redness status.

| Grade | Description Criteria |
|:---:|:---:|
| 1 | No |
| 2 | Slight recognition |
| 3 | Perceptible but not clearly identified light pink |
| 4 | Clear red |
| 5 | Dark red or purple with epidermal damage |

*2.9. Statistics*

The experiments were independently performed three times. All data are expressed as means ± standard deviation. Statistical analysis was conducted using one-way analysis of variance. Values of * $p < 0.05$ and ** $p < 0.01$ were considered significant.

## 3. Results

*3.1. E. cava Extract Protects against Pollutant-Induced Cytotoxicity*

*E. cava* extract was used in vitro to evaluate its protective properties against SRM-1649b, an urban dust substitute. We first conducted cell viability assays to determine whether urban dust can influence cell morphology and cell survival of HFDPC. After treatment with SRM-1649b, we found morphological changes reflected by cell shrinkage, rounding and loss of attachment to the well, which are indicative of apoptosis-inducible features. Also, we found a remarkable dose-dependent reduction in cell number (data not shown). From this data, we chose 200 μg/cm² as an appropriate dosage of SRM-1649b to perform further experiments, as it shows significant changes in cell viability compared to non-treated controls. When the cells were co-incubated with *E. cava* extract for 72 h, we observed a reduction in damage from SRM-1649b and increased the number of cells in a dose dependent manner (Figure 1A). Cell viability was improved by 10.0%, 11.7% and 16.8%, respectively in *E. cava* extract-treated cells against cells that were exposed to SRM-1649b only (Figure 1B).

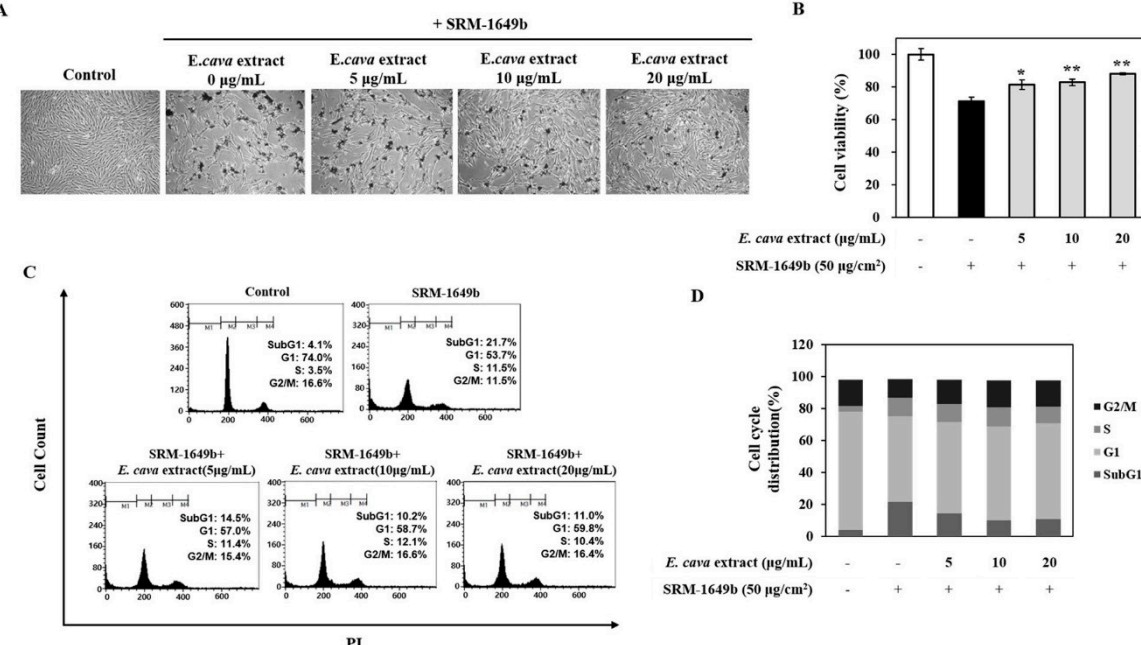

**Figure 1.** SRM-1649b-induced cytotoxicity was reduced by treatment with *E. cava* extract. Cell images of HFDPC with *E. cava* extract added after SRM-1649b treatment (**A**). Quantified viability (**B**). The apoptotic effect induced by SRM-1649b was rescued by *E. cava* extract and it was confirmed in cell cycle analysis using flow cytometry (**C**). Each cell cycle phase is shown graphically (**D**). Data are representative of three independent experiments and values are expressed as mean ± S.D (* $p < 0.05$, ** $p < 0.01$).

In addition, we investigated cell cycle distribution using flow cytometry and asked whether SRM-1649b can induce the sub-G1 peak which is a key feature of apoptosis (Figure 1C) and the data shown graphically (Figure 1D). We confirmed 100 μg/cm² of SRM-1649b can induce a 20% increase in cells in subG1 phase. However, *E. cava* extract was able to reduce the proportion of SRM-1649b-induced subG1 phase cells. These results indicate that *E. cava* extract can protect from SRM-1649b-induced apoptosis in HFDPCs.

*3.2. E. cava Extract Attenuates Pollutant-Induced Loss of Filaggrin and Increased MMP-1 Protein Expression In Vitro*

The reduction or loss of filaggrin expression leads to abnormal barrier formation. A previous study demonstrated that dephosphorylated and proteolysed filaggrin from profilaggrin directly bind to keratin filaments and it can help to strengthen the skin barrier and maintain it from variety of damages [15]. Urban dust, one of the type of pollutants decreased filaggrin expression, which resulted in barrier dysfunction [11]. To determine whether *E. cava* extract protects skin barrier from urban dust, we carried out western blotting to evaluate the protein expression of filaggrin monomers. As shown in Figure 2A, filaggrin monomers expression was decreased by urban dust. Co-treatment with *E. cava* extracts restored expression of the active state of filaggrin monomers. Protein expression is quantified in Figure 2B.

MMP-1 is one of the major skin aging factors induced by pollutants and in this study, we observed that MMP-1 was increased by incubation with SRM-1649b. This change in MMP-1 expression was diminished by *E. cava* extract (Figure 2C). Similar to filaggrin, it became normalized as depicted in the graph (Figure 2D). Based on these results, we conclude *E. cava* extract can protect the skin barrier against urban dust.

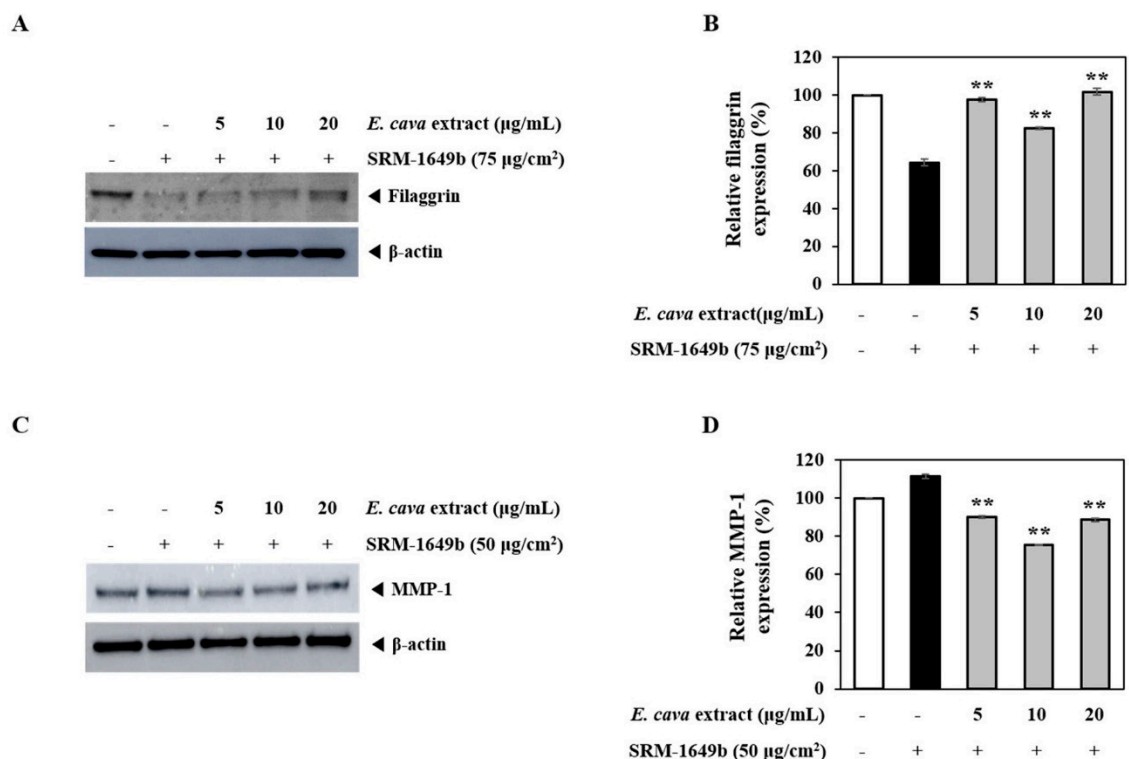

**Figure 2.** Protein expression of filaggrin and MMP-1 in HFDPC after co-treatment with SRM-1649b and *E. cava* extract. Western blot showing filaggrin expression levels when treated with SRM-1649b and *E. cava* extract (**A**). Quantification of filaggrin western blots (**B**). Protein expression by western blot of MMP-1 in co-treatments with SRM-1649b and *E. cava* (**C**). Quantification of MMP-1 western blots (**D**). Data are representative of three independent experiments and values are expressed as mean ±S.D (* $p < 0.05$, ** $p < 0.01$).

*3.3. E. cava Extract Attenuates the Pollutant-Induced Loss of Filaggrin and Increase in MMP-1 Protein Expression Ex-Vivo*

Next, we investigated the protective effect of *E. cava* extract against urban dust in an ex-vivo model. We prepared living skin explant models and applied heavy metals and hydrocarbon mixture-soaked paper disks on each skin explant in the presence or absence of *E. cava* extract. After treatment,

paraffinized, immuno-stained cross sections of the explants were observed by microscopy. In those explants treated with pollutants, there was a moderate reduction in filaggrin expression (green fluorescence). When 0.05% *E. cava* extract was applied, filaggrin expression recovered completely (Figure 3A). Intensity of filaggrin-dependent green fluorescence was quantified by image J (Figure 3B).

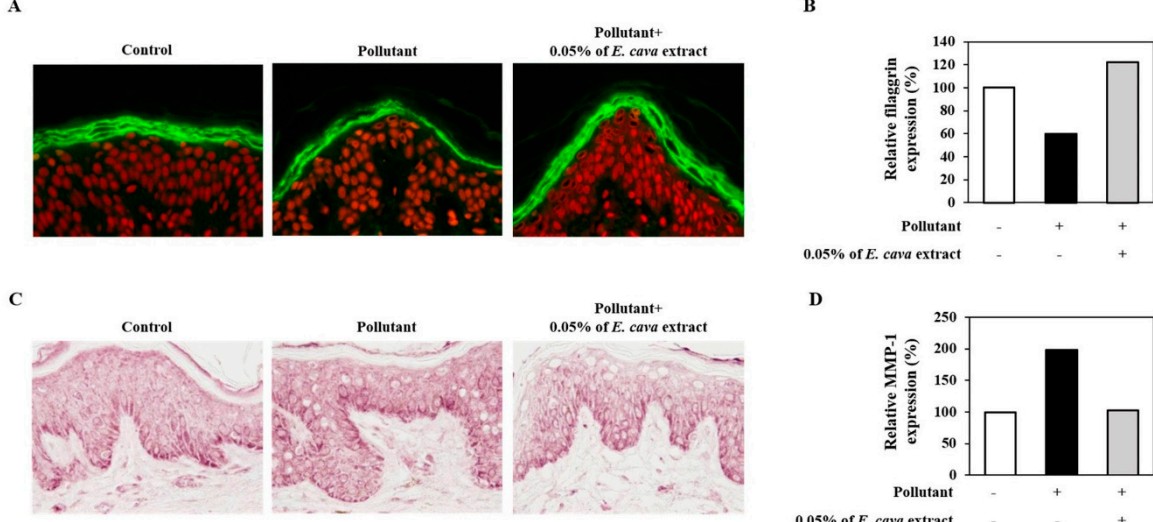

**Figure 3.** Effect of *E. cava* extract on filaggrin and MMP-1 expression conducted in living human skin explant models affected by pollutants. A mixture of heavy metals and hydrocarbon induced loss of filaggrin which was recovered by treatment with *E. cava* extract confirmed in immunostaining microscopic images (**A**) and quantified in (**B**). Pollutant induced expression of MMP-1 was confirmed under the same conditions. Expression of MMP-1 was significantly decreased by *E. cava* extract (**C**) and quantified in graph (**D**). All staining results were evaluated using image J program.

Similarly, MMP-1 was increased only in the pollutant-treated group and *E. cava* extract ameliorates urban dust-induced MMP-1 expression (Figure 3C). Overall staining for MMP-1 was quantified using image J (Figure 3D).

*3.4. Analysis of Scalp Scale, Redness and Hydration*

Clinical testing was conducted to ascertain whether *E. cava* extract has scalp soothing and hydrating effects. Volunteers with scalp scale and redness, used shampoo containing *E. cava* extract for 4 weeks and scalp scale, skin redness and hydration value were measured at baseline and after 4 weeks treatment. Shampoo containing 0.02% *E. cava* extract reduced the average score of the specific area when assessed by two qualified researchers. It was photographed and assessed according to standard criterion. These results showed that scalp scale and redness was decreased significantly after 4 weeks compared to baseline (Figure 4A,B) and demonstrated that *E. cava extract* has scalp soothing effect.

In addition, skin hydration of the frontal and parietal regions was significantly increased compared to baseline (Figure 4C). The change from baseline for the frontal region was 15.5% and that of parietal region was 15.8%. These were statistically significant differences (* $p < 0.05$).

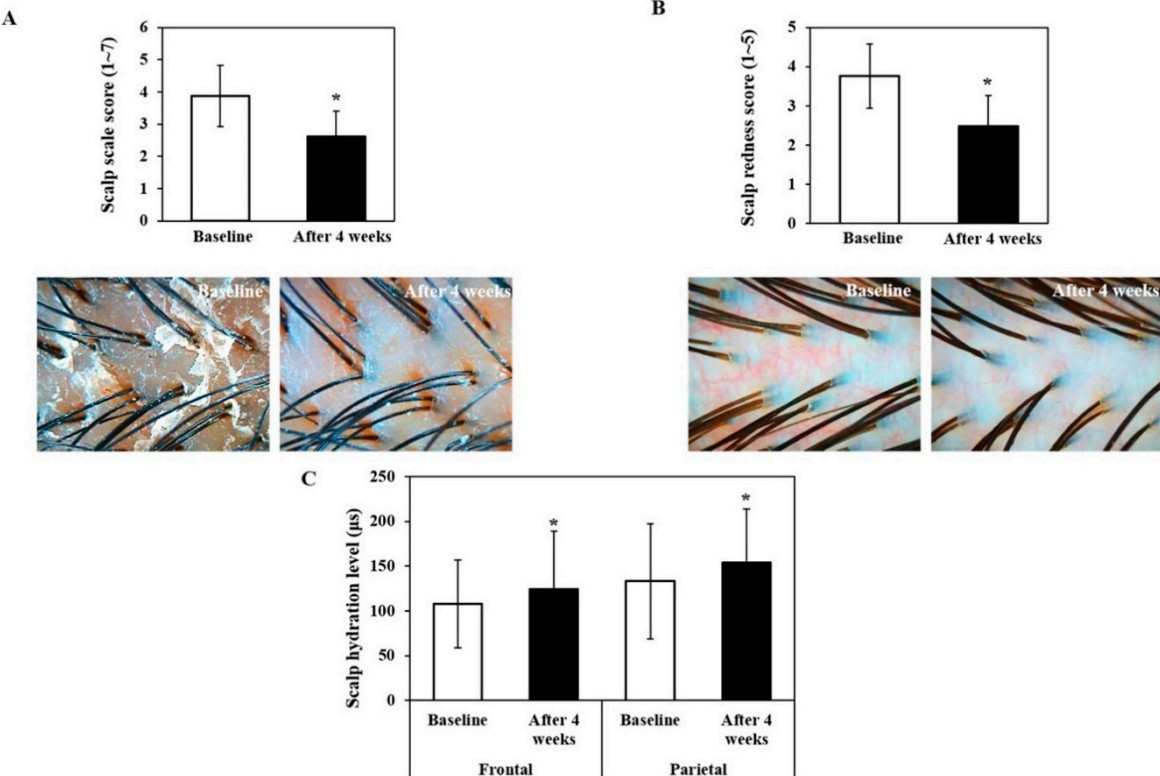

**Figure 4.** Clinical study using 0.02% *E. cava* extract containing shampoo. Female volunteers used *E. cava* extract containing shampoo for 4 weeks to reveal its protective effect on human scalp. The scalp scale and redness were photographed using Aroma TS® and assessed by two qualified researchers according to standard criterion (**A**,**B**). The skin hydration on frontal and parietal region was measured by conductance measurement (**C**). Statistical analysis of the clinical test was conducted using the SPSS (R) software program (IBM).

## 4. Discussion

The scalp is exposed to various irritants such as detergents, microorganisms, dry climate and urban dust [16], and irritation often results from residual detergents and dysregulation of microbiota [17]. Scalp dandruff is primarily attributed to Malassezia yeasts, which are known to provoke upregulation of inflammatory mediators such as IL-8, are chemotactic to neutrophils and irritating fatty acid [18]. Urban dust is one of the biggest threats to human health in many industrializing countries. Oxidative stress and AhR activity are important pathophysiological mechanisms that result from pollutant-induced damage. Exposure of skin to air pollutants is closely related to skin aging and inflammatory responses such as wrinkles formation, hyperpigmentation, atopic dermatitis, and acne [19]. The effect of pollutants on the scalp and hair follicles has been poorly understood. In female pattern hair loss, AhR activation (a major pathway of pollutant-induced damage) was observed in miniaturized hair follicles. In addition, air pollutants, especially environmental cigarette smoke, induce hair loss and premature grey hair in mice [20]. Based on those results, we hypothesized that pollutants can exert adverse effects on HFDPC which play a critical role in hair growth. Our results show that urban dust induces apoptosis in HFDPC, however *E. cava* extract prevented these apoptotic effects. The detailed mechanism by which *E. cava* extract protects HFDPCS needs to be investigated further.

The scalp is an epithelium that generates an effective barrier—the stratum corneum. Filaggrin is vital component of this barrier and exists as assembled pro-filaggrin, bound to keratin fibers in epithelial cells [21]. Filaggrin down-regulation can impair the skin barrier and we confirmed that urban dust induces the downregulation of filaggrin in keratinocytes and in an *ex-vivo* model, as expected. *E. cava* extract then attenuated the reduction in filaggrin levels induced by pollutants. MMP-1 expression is

upregulated by basal keratinocytes in any condition where the epidermis and basement membrane are disrupted [22]. We also observed that MMP-1 expression was upregulated by urban-dust in keratinocytes and in the *ex-vivo* model, however *E. cava* extract decreased this pollutant-induced MMP-1 expression. These results indicate that *E. cava* extract can protect against skin barrier disruption by pollutants.

Urban dust has been reported to cause skin inflammation through upregulation of inflammatory mediators such as ROS, COX, $PGE_2$ and TNF-$\alpha$. *E. cava* conversely, is known to have anti-inflammatory efficacy in various conditions [23,24]. A recent study demonstrated anti-inflammatory effects of *E. cava* against airborne particulate matter (PM), as it was able to suppress PM-induced COX and $PGE_2$ production in keratinocytes [24]. We also confirmed that *E. cava* exerts inhibitory effects on LPS-induced NO production and IL-4-induced eotaxin-1/CCL11 (Figure A1). Based on these results *E. cava* extract may act as an inhibitory material in immune responses to external stimuli.

To confirm whether *E. cava* extract has scalp soothing and hydrating effect on volunteers who have scalp scale and redness, clinical trial was performed. The use of shampoo containing 0.02% of *E. cava* extract for 4 weeks reduced the scalp scale and redness and skin hydration was significantly increased.

## 5. Conclusions

In this study, we have revealed that *E. cava* extract may be a promising candidate for the protection of scalp through preventing urban dust-induced cytotoxic effects and skin barrier dysfunction in keratinocytes and skin explant. Furthermore *E. cava* exhibited scalp soothing effects by increasing hydration and decreasing scalp scale and redness in a clinical trial.

**Author Contributions:** H.K. and E.J. conceived and designed the experiments; H.K. and H.W. performed the experiments; H.K., H.W. and S.S. analyzed the data and interpreted data; D.P. contributed reagent/materials/analysis tools; H.K. wrote the paper and E.J. corrected the paper. All authors read the manuscript and approved the final manuscript.

**Funding:** This research was funded by Ministry of Trade, Industry and Energy, grant number R0002895.

**Acknowledgments:** This study was supported by a grant (R0002895) funded by the Ministry of Trade, Industry and Energy, Republic of Korea.

**Conflicts of Interest:** The authors declare no conflict of interest.

## Appendix A

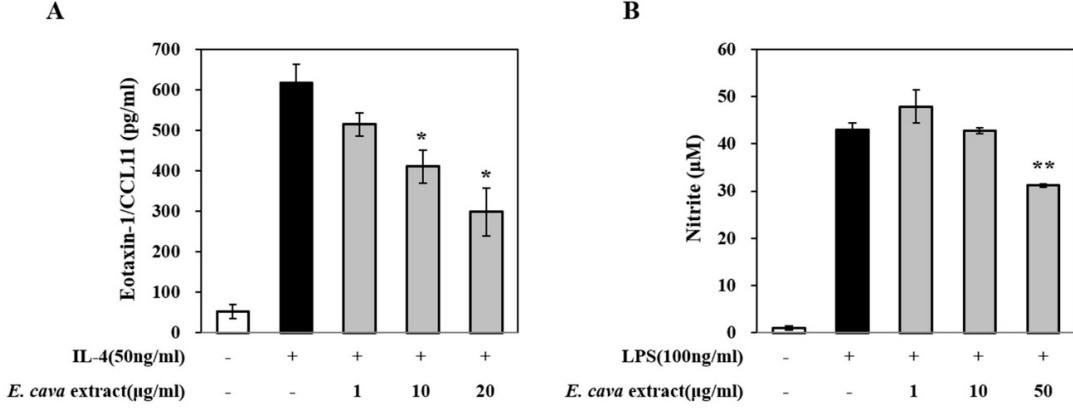

**Figure A1.** Effect of *E. cava* extract on inflammation factors expression induced by pollution stress. NIH/3T3 cells were pretreated with *E. cava* extract, then further incubated with IL-4 (50 ng/mL). Culture cell medium was collected, and the release of eotaxin-1/CCL11 was measured by ELISA (**A**). RAW264.7 cells were pretreated with *E. cava* extract, and then stimulated with lipopolysaccharide (LPS, 100 ng/mL). Culture cell medium was collected, and the release of NO was measured by Griess assay (**B**). Each graph means average value of each data and standard variation was shown as error range. The experiment was conducted in triplicate.

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
