# Peer review of "The Potential Application of Ecklonia cava Extract in Scalp Protection"

_cosmetics, doi:10.3390/cosmetics7010009_

Round 1

Reviewer 1 Report

I attached the comments (Word file).

Author Response

Major points;

(1) E. cava extract contains a lot of components. The authors should speculate the main component showing the improvement of scalp scale and redness.

Answer: In attached reference, keratinocytes were exposed to airbone particulate matter in the presence or absence of dieckol. In result, dieckol attenuated PGE2 production and the gene expression of COX-1, COX-2, and mPGES-1 stimulated by particulate matter. We are predicting that these anti-inflammatory efficacy of dieckol, the component of e. cava extract, has resulted in improved scalp condition and reduced redness.

References:

Ha, J.W.; Song, H.; Hong, S.S.; Boo, Y.C. Marine Alga Ecklonia cava Extract and Dieckol Attenuate Prostaglandin E2 Production in HaCaT Keratinocytes Exposed to Airborne Particulate Matter. Antioxidants. 2019, 21(8).

(2) In Figure 1, the treatment with e. cava extract protected the cells from urban dust induced apoptosis. What kind of signal transduction is stimulated by treatment with E. cava extract?

Answer: Phloroglucinol, a kinds of phlorotannins, is another components of e. cava, is reported to protect cells from oxidative stress-induced DNA damage and apoptosis through activating the Nrf2/HO-1 signaling pathway. In attached research, phloroglucinol suppressed H2O2-induced apoptosis in HaCaT keratinocytes and it also restored H2O2-induced alteration of the apoptosis regulatory genes such as Bcl-2, Bax, caspase-9, caspase-3 and PARP. Also many studies have shown that particulate matter able to induce reactive oxygen species (ROS). Collectively, we are predicting that urban dust induced apoptosis is regulated by this anti-oxidative effect by phloroglucinol via Nrf2/HO-1 signaling pathway.

References:

Park, C.; Cha, H.J.; Hong, S.H.; Kim, G.Y.; Kim, S.; Kim, H.S.; Kim, B.W.; Jeon, Y.J.; Choi, Y.H. Protective Effect of Phloroglucinol on Oxidative Stress-Induced DNA Damage and Apoptosis through Activation of the Nrf2/HO-1 Signaling Pathway in HaCaT Human Keratinocytes. Mar Drugs. 2019, 13. 17(4).

(3) In Figure 2, the treatment with E. cava extract showed decrease in MMP-1 and increase in filaggrin expressions. To control the expression of these proteins, do several components in E. cava extract act? Or does a definite component act?

Answer: Previous studies have been reported that urban particulate matter upregulate COX-2 expression and increase PGE2 production in human keratinocytes, resulting in down-regulation of filaggrin and may lead to skin barrier dysfunction.

Also as answered in question (1), dieckol, a polyphenols from e. cava has been explored for its ability to alleviate the inflammation effect such as gene expression of COX-1, COX-2, and PGE2 production caused by urban particulate matter. Through down regulation of expression of COX-1, we thought that dieckol as helping to protect expression of filaggrin and maintain skin barriers.

The other study demonstrated the inhibitory effect of dieckol on MMP-1 expression in human dermal fibroblast.

Based on these evidence, we are expecting dieckol as an important component in protective effect of e. cava, but it has not been confirmed whether another substance exists.

References:

Ha, J.W.; Song, H.; Hong, S.S.; Boo, Y.C. Marine Alga Ecklonia cava Extract and Dieckol Attenuate Prostaglandin E2 Production in HaCaT Keratinocytes Exposed to Airborne Particulate Matter. Antioxidants. 2019, 21(8).

Lee, C.W.; Lin, ZC.; Hu, SC.; Chiang, Y.C.; Hsu, LF.; Lin, Y.C.; Lee, I.T.; Tsai, M.H.; Fang, J.Y. Urban particulate matter down-regulates filaggrin via COX2 expression/PGE2 production leading to skin barrier dysfunction. Scientific Reports 2016, 6, 27995.

Minor points;

(1) In page 2 (line 6), “in in miniaturized” should be replaced with “in miniaturized”.

Answer: It has been changed as you mentioned.

(2) In page 3 (line 16), what is “BEM” culture medium? The authors should describe the full name of “BEM”.

Answer: It has been changed in full name.

Reviewer 2 Report

The article is well written 

Author Response

Some sentences have been edited.

Thank you for your review.

Reviewer 3 Report

Good job! Congratulations!

Author Response

Thank you for your review.